# Activation of MyD88-Dependent TLR Signaling Modulates Immune Response of the Mouse Heart during *Pasteurella multocida* Infection

**DOI:** 10.3390/microorganisms11020400

**Published:** 2023-02-04

**Authors:** Qiaoyu Fu, Junming Jiang, Xubo Li, Zhe Zhai, Xuemei Wang, Chongrui Li, Qiaoling Chen, Churiga Man, Li Du, Fengyang Wang, Si Chen

**Affiliations:** Hainan Key Laboratory of Tropical Animal Reproduction & Breeding and Epidemic Disease Research, Animal Genetic Engineering Key Laboratory of Haikou, School of Animal Science and Technology, Hainan University, Haikou 570228, China

**Keywords:** zoonotic pathogen, *Pasteurella multocida*, immune response, heart infection, MyD88

## Abstract

*Pasteurella multocida* (*P. multocida*) is an important zoonotic pathogen. In addition to lung lesions, necropsies have revealed macroscopic lesions in the heart in clinical cases. However, most previous studies focused on lung lesions while ignoring heart lesions. Therefore, to investigate the immune response of the *P. multocida*-infected heart, two murine infection models were established by using *P. multocida* serotype A (Pm HN02) and D (Pm HN01) strains. Histopathological examination revealed heterogeneous inflammatory responses, including immune cell infiltration in the epicardial and myocardial areas of the heart. Transcriptome sequencing was performed on infected cardiac tissues. To explore the traits of immune responses, we performed the functional enrichment analysis of differentially expressed genes, gene set enrichment analysis and gene set variation analysis. The results showed that the innate immune pathways were significantly regulated in both groups, including the NOD-like receptor signaling pathway, the complement and coagulation cascade and cytokine–cytokine receptor interaction. The Toll-like receptor signaling pathway was only significantly activated in the Pm HN02 group. For the Pm HN02 group, immunohistochemistry analysis further verified the significant upregulation of the hub component MyD88 at the protein level. In conclusion, this study reveals critical pathways for host heart recognition and defense against *P. multocida* serotypes A and D. Moreover, MyD88 was upregulated by *P. multocida* serotype A in the heart, providing a theoretical basis for future prevention, diagnosis and treatment research.

## 1. Introduction

The Hainan black goat is an indigenous goat that evolved under tropical climatic conditions of high humidity and temperature [1]. It is also an economically important livestock species for local animal husbandry. Goat hemorrhagic septicemia (HS) is an acutely fatal infectious disease resulting from *Pasteurella multocida* (*P. multocida*) [2], causing serious economic losses in many countries. Based on its capsular antigen, *P. multocida* is grouped into five serotypes: A, B, D, E and F [3]. Goats are the preferred hosts for *P. multocida* serotype A and D strains [4]. The capsular antigen of serotype A, mainly composed of hyaluronic acid, promotes the colonization of *P. multocida* in the mucosa of the lower respiratory tract [5]. Thus, the *P. multocida* strain of serotype A usually causes respiratory diseases. The main component of the serotype D capsule is heparosan [6]. The *P. multocida* strain of serotype D is usually associated with HS and pneumonia [7]. Our previous study demonstrated that *P. multocida* strains of serotypes A (Pm HN02) and D (Pm HN01) led to different cellular morphologies in goat bronchial epithelial cells [8]. Whether the host immune responses induced by these two strains are different has not been reported.

Ordinarily, *P. multocida* results in pneumonia and HS [9]. There are many studies regarding the lung [10,11] but fewer about the heart, despite reports of heart lesions caused by *P. multocida* [12]. Weise et al. demonstrated that *P. multocida* toxin aggravated cardiac hypertrophy and fibrosis in mice [13]. Clinically, cardiac damage caused by *P. multocida* also includes hemorrhages in subepicardial and subserosal areas [14], endocarditis [15], chronic fibrous pericarditis [16] and congestion with hemorrhages in necropsy hearts [17,18]. Moreover, Pors, Susanne E. detected and cultivated *P. multocida* from the pericardial sacs of 40 pigs [16]. The *P. multocida* burden was detected in mice heart after 12 h of *P. multocida* infection [19]. Bacterial virulence factors (LPS, etc.) are known to induce cardiac inflammation and myocardial cell death [20]. In conclusion, *P. multocida* invades the heart and causes pathological changes. We therefore focus on the immune mechanisms behind pathological changes in the heart caused by Pm HN01 and Pm HN02.

Previous study indicated that the mouse, being highly sensitive and susceptible to *P. multocida*, could serve as a useful model for pathogenesis studies, such as challenge studies, including for the development of vaccines against HS [21]. Therefore, we conducted subsequent in vivo experiments in mice infected with *P. multocida*. In order to systematically evaluate the pathological damage to the heart caused by the two strains and explore the immune response during the infection, Pm HN01 and Pm HN02 strains preserved in our laboratory were used to construct models of *P. multocida* infection by intraperitoneal injection. Firstly, a histopathological examination of heart tissues was performed. Subsequently, the analysis of transcriptome sequencing data was used to explore the immune response of mouse heart tissues. Additionally, functional enrichment analyses were performed using Gene Ontology (GO), Kyoto Encyclopedia of Genes and Genomes (KEGG), gene set enrichment analysis (GSEA) and gene set variation analysis (GSVA). Finally, quantitative real-time PCR (RT-qPCR) was performed to check the expression of 10 immune-related differentially expressed genes (DEGs). One of the hub genes, myeloid differentiation factor 88 (*Myd88)*, was identified by immunohistochemistry (IHC) staining at the protein level. This paper reveals the critical pathways of host heart recognition and defense against *P. multocida* serotypes A and D. These pathways and genes provide potential targets for therapeutic interventions of Pasteurellosis.

## 2. Materials and Methods

### 2.1. Animals

A total of 24 eight-week-old, specific-pathogen-free (SPF), male KM mice (weight: 36 ± 2 g) were purchased from HUNAN SJA LABORATORY ANIMAL CO., LTD (Hunan, China). The mice were fed sterilized food and water for 3 days and were randomly divided into 3 groups: the control group (control group, *n* = 9), the infected group challenged with Pm HN01 (Pm HN01 group, *n* = 6) and the infected group challenged with Pm HN02 (Pm HN02 group, *n* = 9). Of the 24 mice, 9 mice (3 in each group) were examined using histological staining, another 9 mice (3 in each group) were used for transcriptome sequencing, and the remaining 6 mice (3 in the control group and 3 in the Pm HN02 group) were used for immunohistochemistry staining. Accordingly, there were 3 biological replicates for each experiment.

### 2.2. P. multocida Challenge Experiments

The bacterial strains used in this study were previously isolated in our laboratory. Specifically, Pm HN01 (GenBank Accession No. Cp037861.1) and Pm HN02 (GenBank Accession No. Cp037865.1) were isolated from lung tissues of a Hainan black goat and a sheep, respectively. The bacterial strains were routinely cultured in tryptic soy broth (Hopebio, Qingdao, China) containing 5% (v/v) newborn bovine serum (Sijiqing, Beijing, China). According to our previous research, the challenge doses for each mouse in the Pm HN01 and Pm HN02 groups were 1.58 × 10^6^ Colony-Forming Units (CFU) and 1.58 × 10^7^ CFU at 100 µL by intraperitoneal injection, with an equal volume of sterile phosphate-buffered saline (PBS) for the control group. Moribund mice were immediately euthanized by cervical dislocation. The heart tissues were collected and stored in liquid nitrogen for RNA extraction to perform transcriptome sequencing or in 4% paraformaldehyde for the other two experiment.

### 2.3. Histopathological Examination

Hematoxylin and eosin (HE) staining was conducted according to routine protocols [22]. Briefly, paraffin-embedded hearts were cut into 4 μm thick sections using LEICA RM2016 (LEICA, Wetzlar, Germany). Next, sections were dewaxed in xylene, dehydrated using gradient alcohol and stained by the HE dye solution set (Servicebio, Wuhan, China). Finally, the tissue was sealed with neutral gum (SCRC, Shanghai, China) and directly observed with a microscope (Nikon, Tokyo, Japan).

### 2.4. RNA Extraction and Transcriptome Sequencing

Total RNA was extracted from heart tissues using the TRIzol (TIANGEN, Beijing, China). Subsequently, the RNA concentration and purity were determined by NanoDrop 2000 (Thermo Fisher Scientific, Waltham, MA, USA). The integrity of RNA was assessed using the RNA Nano 6000 Assay Kit of the Agilent Bioanalyzer 2100 system (Agilent Technologies, Santa Clara, CA, USA). According to the manufacturer’s procedure, the sequencing library was generated using the NEBNext UltraTM RNA Library Prep Kit for Illumina (NEB, Ipswich, MA, USA) and evaluated for library quality on the Agilent Bioanalyzer 2100 system. The libraries were sequenced on the Illumina platform. The raw data were filtered to obtain clean data. HISAT2 (Version 2.0) was used to map the reference genome (GenBank Assembly ID: GCA_000001635.8). Fragments Per Kilobase of transcript per Million fragments mapped (FPKM) reads were used to quantify gene expression levels obtained using StringTie (Version 1.3.0). The repeatability among samples was evaluated by Spearman’s correlation coefficient.

### 2.5. Bioinformatics Analysis of Sequence Data

Deseq2 was used to analyze DEGs by Fold Change (FC) and *p* value between the infected group and the control group. The screening conditions were |log_2_ FC| ≥ 1.5 and *p* < 0.05. The DEGs were analyzed using DiVenn software [23] (Version 2.0), GO and KEGG databases. Additionally, the KEGG database was predefined as the background gene set. GSEA [24] was used for all expressed genes to evaluate the expression of pathways. Thresholds of *p* < 0.05, False Discovery Rate < 0.25 and | Normalized Enrichment Score | > 1 were the screening criteria. Subsequently, GSVA [25] was used to further screen immune pathways with significant expression differences according to the threshold of |FC| > 1.

### 2.6. Quantitative Real-Time PCR

Based on the transcriptome data, ten immune- and inflammation-related DEGs were selected for RT-qPCR verification. Specific primers were designed according to the reference sequences in Genebank with Primer (Version 5.0), and the primer sequences are listed in Appendix A. The total RNA was reverse-transcribed into cDNA using FastKing gDNA Dispelling RT SuperMix (TIANGEN, Beijing, China). RT-qPCR was performed on QuantStudio™ 6 Flex (ABI, Los Angeles, CA, USA) with SuperReal PreMix Plus (SYBR Green) (TIANGEN, Beijing, China). Relative expression analysis was performed using the formula 2^−ΔΔCt^ [ΔΔCt = (Ct of the target gene – Ct of the *β-actin* gene) in the treatment group – (Ct of the target gene – Ct of the *β-actin* gene) in the control group].

### 2.7. Immunohistochemistry Staining

Immunohistochemistry staining was conducted using 4 μm paraffin-embedded heart tissue (*n* = 6) sections. The primary antibodies against MyD88 (GB11269, Wuhan Servicebio Biotechnology Co., Ltd., Wuhan, China) were diluted 1:400 and then incubated at 4 °C overnight in a humidified container. After three washes with PBS, we added horseradish peroxidase (HRP)-labeled Goat Anti-Rabbit IgG (GB23303, Wuhan Servicebio Biotechnology Co., Ltd., Wuhan, China), which was diluted 1:200 and incubated at room temperature for 50 min. After three more washings with PBS, a chromogenic solution was added, and chromogenesis was terminated with tap water. Nuclei were then stained with hematoxylin. Finally, the slices were dehydrated and sealed, and the results were interpreted under a white-light microscope. Twelve fields of view were randomly selected from 400-fold fields (Appendix A). The quantification of staining intensity was analyzed using the Image Pro Plus software (Version 6.0).

### 2.8. Statistical Analysis

All data are depicted as means ± SEM, and error bars in graphs indicate SEM. Results are representative of three independent experiments. Statistical significance was analyzed using using two-tailed unpaired Student’s *t*-test in GraphPad Prism (Version 8.0).

## 3. Results

### 3.1. Pm HN01 and Pm HN02 Cause Different Pathological Changes in the Heart

The histopathological examination was performed to evaluate the effect of *P. multocida* infection on the murine heart. In the control group, cardiomyocytes were neatly arranged (Figure 1A). The nuclei were oval in shape and centrally located. Neither tissue degeneration nor inflammatory cell infiltration was observed. In the Pm HN01 group, myocardial hyperemia can be seen. Myocardial fibers were disorderly arranged, and the intercellular space was widened. The heart tissue showed extensive degeneration and necrosis. Inflammatory cells were aggregated under the epicardium. Multiple focal inflammatory cell aggregates were observed in the myocardium (Figure 1B). As for the Pm HN02 group, myocardial hyperemia and myocardial disarray were evident. The intercellular space was slightly widened. Additionally, mild degeneration and necrosis were observed in the tissue, and inflammatory cell infiltrates were scattered in the myocardium (Figure 1C). In contrast, mice challenged with Pm HN01 presented more serious pathological damage.

### 3.2. Pm HN01 and Pm HN02 Induce Unique and Common DEGs

Because of the immune cells observed in the histopathological examination, we intended to explore the immune pathways therein. Hence, transcriptome sequencing was performed on the murine heart. The quality of sequencing data was assessed. The transcript abundance was estimated based on FPKM, and all samples showed more upregulated transcripts than downregulated transcripts (Appendix A). The Spearman correlation matrix analysis of all nine samples showed that the measurements in each group were highly consistent and reproducible (Appendix A). DEGs were screened from transcriptome sequencing data to further determine the effect of *P. multocida* on host cardiac tissues.

The results of DEG analysis in DiVenn software identified 2243 common DEGs in the differential expression gene sets of the Pm HN01 and Pm HN02 groups (Figure 2). Among them, the number of upregulated and downregulated genes was 1242 and 998, respectively. Additionally, the expression of these genes in both groups showed similar tendencies. These results indicated that the host response to Pm HN01 exhibited certain similarities to that to Pm HN02. In addition, the Pm HN01 group had 1107 unique DEGs, while the Pm HN02 group had 849. These unique DEGs imply the specificity of the two strains to the host.

### 3.3. Common DEGs Regulated by Pm HN01 and Pm HN02 Are Mainly Related to Innate Immunity

To explore the interactions between the host and the two different *P. multocida*, GO enrichment analysis was implemented on the common DEGs in the Pm HN02 and Pm HN01 groups (Figure 3A). In the biological process (BP) category, DEGs were mainly enriched in the regulation of biological process and regulation of response to stimulus (Figure 3B). In the molecular function (MF) category, binding was the most significant function, such as protein binding and cell adhesion molecule binding (Figure 3C). In the cellular component (CC) category, the intracellular part was the top GO term (Figure 3D). The DEGs were further annotated from the KEGG database to assess the critical signaling pathways (Figure 4). A total of 2709 DEGs were enriched in KEGG pathways involved in metabolism (global and overview maps), environmental information processing (signal transduction), organismal systems (immune system) and human diseases (infectious diseases) (Figure 4A). The top 20 KEGG pathways were mainly enriched in immune-related pathways, such as the nucleotide-binding oligomerization domain (NOD)-like receptor signaling pathway, the complement and coagulation cascade and cytokine–cytokine receptor interaction (Figure 4B).

### 3.4. Unique DEGs Caused by Pm HN01 and Pm HN02 Are Functionally Distinct

Unique DEGs from the Pm HN01 and Pm HN02 groups were selected according to the DiVenn analysis (Figure 2) for GO and KEGG analyses (Figure 5). Unique DEGs in the Pm HN01 group were annotated to 58 GO terms, including 27 BPs, 13 CCs and 17 MFs. Comparatively, unique DEGs in the Pm HN02 group were annotated to 53 GO terms, including 26 BPs, 11 CCs and 15 MFs. Subsequently, the top 20 GO terms were screened for further analysis based on the *p* values. The results showed that unique DEGs in the Pm HN01 group were mainly enriched in protein binding (Figure 5A), while cytoplasm was the most significant GO term in the Pm HN02 group (Figure 5B).

This study focused on the immune-related pathways enriched in KEGG. Among them, unique DEGs in the Pm HN01 group were mostly enriched in the mitogen-activated protein kinase (MAPK) signaling pathway, followed by the chemokine signaling pathway (Figure 5C). The unique DEGs in the Pm HN02 group were mostly enriched in endocytosis, adherens junction, bacterial invasion of epithelial cells and the Toll-like receptor signaling pathway (Figure 5D). Therefore, it was speculated that the differences between the pathogenesis of Pm HN01 and Pm HN02 were related to these signaling pathways.

### 3.5. Toll-like Receptor Signaling Pathway Significantly Activated by Pm HN02

GSEA is a powerful analytical method that focuses on gene sets, which share common biological functions, chromosomal locations or regulatory mechanisms. The results of GSEA (Figure 6A,B) revealed that all genes in the Pm HN01 group were enriched in 316 pathways, while those in the Pm HN02 group were enriched in 314 pathways that were also in the Pm HN01 group. Among them, 16 immune-related pathways were found in the DEG enrichment analysis. They were the NOD-like receptor signaling pathway, cell adhesion molecules, MAPK signaling pathway, Nuclear Factor κB (NF-κB) signaling pathway, phagosome, Janus kinases (JAK) and signal transducers and activators of transcription (STAT) pathways (JAK-STAT) signaling pathway, Tumor Necrosis Factor (TNF) signaling pathway, chemokine signaling pathway, Toll-like receptor signaling pathway, complement and coagulation cascades, Interleukin (IL)-17 signaling pathway, cytokine–cytokine receptor interaction, leukocyte transendothelial migration, B-cell receptor signaling pathway, ferroptosis and C-type lectin receptor signaling pathway. Meanwhile, these pathways were activated in both the Pm HN01 and Pm HN02 groups.

GSVA is a method that builds on GSEA and is better than GSEA for characterizing pathways from the data obtained. GSVA can be used to compare differences in pathways between control and infected groups to better determine the activity of each pathway. According to the results of GSVA, the pathways shared by the Pm HN01 and Pm HN02 groups were the NOD-like receptor signaling pathway, the complement and coagulation cascade and cytokine–cytokine receptor interaction. Although there was no marked activation of the Toll-like receptor signaling pathway in the Pm HN01 group, it was significantly activated in the Pm HN02 group (Figure 7A,B).

### 3.6. Pm HN02 Induced the Activation of Myd88 at Transcription and Protein Levels

Five DEGs (*Junb*, *Iigp1*, *Gadd45g*, *C3* and *Il4ra*) in common pathways, including the TNF and MAPK signaling pathway, the complement and coagulation cascade and cytokine–cytokine receptor interaction, were verified by RT-qPCR to check the accuracy of the transcriptome sequencing results. RT-qPCR verification demonstrated consistent results with RNA-seq (Figure 8). Due to the different activation statuses of the Toll-like receptor signaling pathway in the two challenge groups, we selected a hub gene (*Myd88*) and other related DEGs in the pathway to determine their expression trends at the transcriptional level. All of the selected genes showed upregulation in RT-qPCR and RNA-seq. Moreover, the MyD88 signal was detected in the cytoplasm by immunohistochemistry (Figure 9A). We found that the levels of *Myd88* in the hearts of Pm HN02-infected mice were significantly higher than that in control mice (Figure 9B).

## 4. Discussion

Previous reports indicated that *P. multocida* led to pathological changes in cardiac inflammation in heart tissue. Nevertheless, little is known about the interaction between the host heart and *P. multocida*. As previously reported, the mouse is a useful animal model for studying *P. multocida* infection [26,27,28]. We constructed the murine infection model via intraperitoneal injection [29] of Pm HN01 and Pm HN02 strains to explore the immune response in the heart. In order to study the host’s recognition and defense against these two different serotype strains, the two strains were required to maintain the same lethal effect, so different challenge doses were selected for the two strains based on Lethal Dose 50. In this study, *P. multocida*-infected mice showed similar clinical symptoms to those of *P. multocida*-infected Hainan black goats, such as appetite loss, messy fur and depression [30]. Additionally, it showed heterogeneous myocardial damage and myocardial congestion. The histopathological examination revealed the infiltration of immune cells, including monocytes and lymphocytes, in epicardial and myocardial areas. However, the immune responses induced by cardiac injury are still largely unknown. Therefore, we analyzed the immune DEGs and signaling pathways to preliminarily characterize the changes at the transcript level and the similarities and differences in host immune responses to the two strains.

In bacterial myocarditis, numerous immune cells play critical roles [30], such as neutrophils, lymphocytes and macrophages [31]. In the present study, macrophages were detected by histopathological examination. Meanwhile, the expression of genetic markers of macrophages (*Cd14*, *Chil3*, *Cd274*, *Cd163*, *Il-1β*, *Ccl5*, *Ccl2*, *Cd14*, *Cd80*, etc.) was significantly upregulated at the genetic level. Additionally, histopathological examination showed myocardial infarction, which was supported by the transcriptome data. Cardiomyocyte marker genes (*Hand2*, *Vcam1*, *Gja1*, *Actc1*, etc.) were significantly downregulated. Furthermore, the functional investigation indicated that NF-κB inhibitor alpha (*Nfkbia*) [32] inhibited cell proliferation and promoted cell apoptosis. In this study, the expression of the *Nfkbia* gene was significantly increased. Hence, myocardial infarction may involve cardiomyocyte apoptosis [33]. Through damaged endothelial cells, intraperitoneally injected bacteria may have entered the blood circulation, adhering to heart tissue. Subsequently, the bacteria were transported to cardiomyocytes and caused heart injury. To sum up, we speculate that the innate immune system may play an essential role in the *P. multocida*-infected murine model. We next investigated key genes and their related pathways.

As the first line of host defense against bacterial infections [34], the innate immune system mainly relies on pattern recognition receptors (PRRs) [35] to recognize pathogen-associated molecular patterns (PAMPs), such as LPS [36]. Nucleotide-binding oligomerization domain (NOD)-like receptors (NLRs) are intracellular receptors in the PRR family [37]. As a vital member of the NLR family, NOD2 is a sensor of bacterial peptidoglycan (PGN) recognition [38]. Previous studies have shown that the basal expression of *Nod2* is low, which is consistent with our control group results. However, after infection with Pm HN01 and Pm HN02, the expression of *Nod2* was significantly upregulated, which may be attributed to the PGN fragment in the *P. multocida* periplasmic space [39]. *NLRC5*, a member of the NLR family, is a negative regulator of the inflammatory pathway. Its expression level was significantly upregulated after LPS stimulation [40]. The expression of *Nlrc5* was significantly upregulated after PmHN01 and PmHN02 infection in our study, which indicated that the body inhibited the inflammatory response to *P. multocida* infection by enhancing the expression of *Nlrc5*. In summary, during infection with Pm HN01 and Pm HN02, genes related to immune inflammation, such as *Nod2* and *Nlrc5*, were generally upregulated and significantly activated the NOD-like receptor signaling pathway. Taken together, the NOD-like receptor signaling pathway is essential for the recognition of Pm HN01 and Pm HN02.

As downstream pathways of PRRs, complement and coagulation and cytokine–cytokine receptor interaction were significantly activated in this study. Complement and coagulation are pivotal factors contributing to inflammation [41] and the core of innate immune mechanisms against extracellular bacteria. In this study, the complement and coagulation cascade was significantly upregulated in the Pm HN01 and Pm HN02 groups, which was an essential manifestation of host heart tissue participating in *P. multocida* infection. Moreover, we detected that the hub gene (*C3*) of the complement and coagulation cascade was upregulated. Thus, these results further indicate that *P. multocida* activated the complement and coagulation cascade. Inflammation stimulates the high expression of various pro-inflammatory mediators and cytokines in the tissue, which is consistent with the activation of cytokine–cytokine receptor interactions in this study. The increased expression of a cytokine (*Il-1β*) and cytokine receptor (*Il4ra*) was validated by RT-qPCR analysis. According to the histopathological examination, inflammatory cell infiltrates were scattered in the myocardium. IL-1β is the central mediator of inflammation [42] and is crucial for the body’s defense against infection. It is possible that Il-1β acts on macrophages through cell surface receptors and induces the expression of certain chemokines and adhesion molecules, thus recruiting and activating immune cells such as neutrophils. In this study, the expression levels of *Il-1β*, *Il4ra*, *Il-6*, *Myd88*, *Nek7* and *Nlrp3* were significantly upregulated and significantly activated complement and coagulation and cytokine–cytokine receptor interaction, suggesting that they are crucial for host defense mechanisms to both Pm HN01 and Pm HN02.

Toll-like receptors (TLRs) are transmembrane signal transduction receptors, which activate the innate immune response of the host through PAMPs, thus eliminating pathogens and playing an essential role in immune defense [43]. This study found that the Toll-like receptor signaling pathway was significantly upregulated during Pm HN02 infection, which is consistent with our in vitro experiment [8]. Lipoprotein on the surface of bacteria and LPS on the outermost layer of the cell wall are essential virulence factors of *P. multocida* [2]. They are also potent agonists that give rise to inflammatory responses. Among TLRs, TLR2 plays a vital role in the early innate immune response to bacterial infection by initiating the release of pro-inflammatory cytokines and influencing the downstream immune response [44]. LPS triggers the primary immune response and finally humoral immunity through TLR4 on the cell surface [45]. In this study, the expression levels of *Tlr2* and *Tlr4* were significantly upregulated after Pm HN01 and Pm HN02 infection, which was consistent with previous test results of *P. multocida* infection [46,47]. The TLR4-mediated signaling pathway of MyD88 activates NF-κB and initiates the production of a series of inflammatory cytokines, such as IL-6 [48]. In addition, TLR4 activates NLRP3 inflammatory bodies [49] through a non-classical pathway by recognizing extracellular LPS, which is pivotal in the host defense against pathogen infection and the regulation of inflammatory responses. Nek7 directly combines with NLRP3 receptor molecules [50], which promotes the assembly of NLRP3 inflammatory bodies. Moreover, the Toll-like receptor signaling pathway was only significantly upregulated in the Pm HN02 group, and no significant difference was observed in the Pm HN01 group in this study, which implies that the Toll-like receptor signaling pathway was stronger in the Pm HN02 group compared to the Pm HN01 group. Then, we focused on the Toll-like receptor signaling pathway in the Pm HN02 group without the Pm HN01 group. Previous studies demonstrated that LPS recognition results in the activation of the MyD88-dependent signaling pathway [51]. Moreover, the increased expression of *Myd88* (the hub gene of the Toll-like receptor signaling pathway in the Pm HN02 group) was verified at both the transcription and protein levels. It has been suggested that *Pasteurella multocida* toxin-induced G protein signaling regulates TLR4-mediated immune responses [52]. Given the results presented here, we can speculate that the key difference in the host defense against the two strains is the Toll-like receptor signaling pathway. Perhaps Pm HN01 evades the immune system by avoiding recognition by TLRs, thereby reducing the innate immune response. This perhaps explains previous studies that suggested that Pm HN02 is less virulent than Pm HN01 [8]. To demonstrate this, further systematic explorations are needed.

## 5. Conclusions

To our knowledge, this study is the first to reveal vital pathways by which the heart recognizes and defends against *P. multocida* serotypes A and D. In addition, the significant activation of the NOD-like receptor signaling pathway suggests the importance for the heart recognition of *P. multocida*. As downstream pathways of PRRs, the significantly activated complement and coagulation cascade and cytokine–cytokine receptor interaction were crucial for the heart innate immune response during *P. multocida* infection. Moreover, *P. multocida* serotype A significantly activated the Toll-like receptor signaling pathway and upregulated the expression of MyD88 in the heart. These signaling pathways can be used as prognostic and diagnostic markers. Additionally, signaling pathway inhibitors or activators can be used as therapeutic targets. Together, these findings provide a molecular basis for the prevention and control of Pasteurellosis.

## Figures and Tables

**Figure 1 microorganisms-11-00400-f001:**
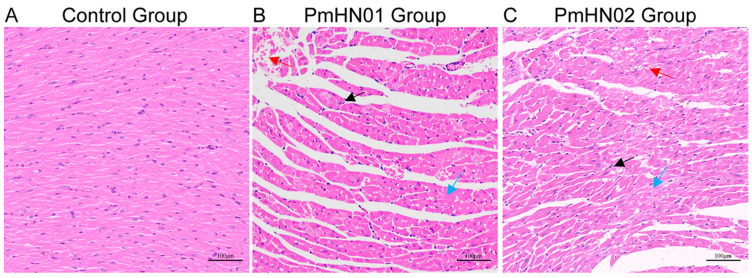
Pm HN01 and Pm HN02 cause different pathological changes in the heart. (**A**–**C**) The results of HE staining of hearts from the 3 groups. The images were magnified by 200× and labeled at 100 µm. The black arrows point to inflammatory cells, the blue arrows point to degeneration and necrosis of myocardial cells and the red arrows point to myocardial congestion.

**Figure 2 microorganisms-11-00400-f002:**
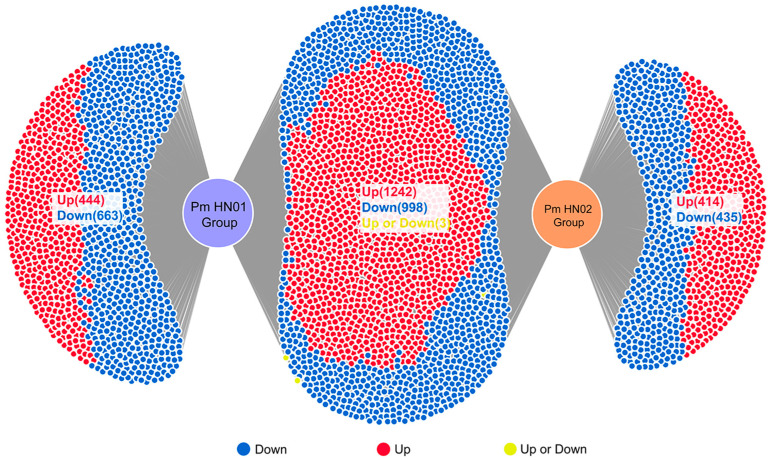
Pm HN01 and Pm HN02 induce unique and common DEGs. The plots represent the unique and common DEGs in different comparisons. Upregulation is indicated in red, while downregulation is indicated in blue.

**Figure 3 microorganisms-11-00400-f003:**
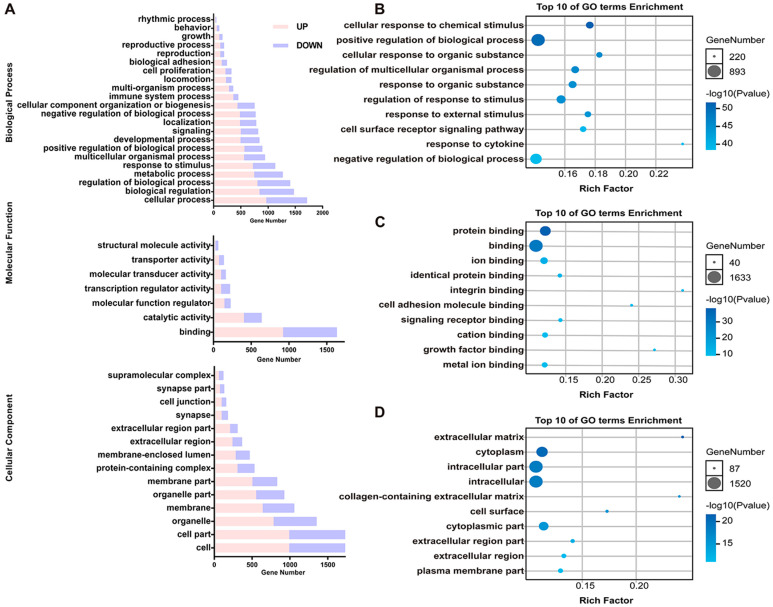
GO enrichment analysis of common DEGs shared by Pm HN01 and Pm HN02 groups. (**A**) The GO enrichment results of DEGs shared by the Pm HN01 and Pm HN02 groups in three categories: biological process (BP), molecular function (MF) and cellular component (CC), and their main level 2 GO terms. The top 10 GO enrichment results for BP (**B**), MF (**C**) and CC (**D**).

**Figure 4 microorganisms-11-00400-f004:**
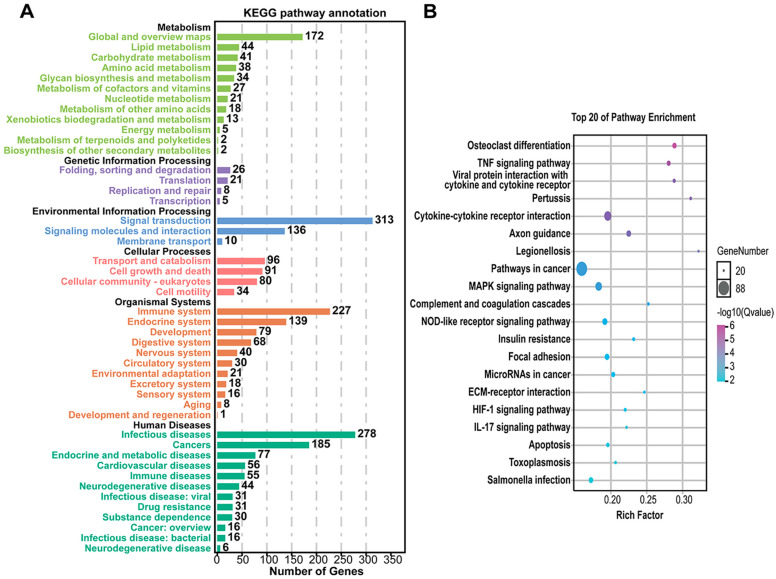
KEGG enrichment analysis of common DEGs shared by Pm HN01 and Pm HN02 groups. (**A**) DEG enrichment in KEGG pathways related to metabolism, cellular processes, environmental information processing, genetic information processing and organismal systems. (**B**) The top 20 KEGG enrichment results.

**Figure 5 microorganisms-11-00400-f005:**
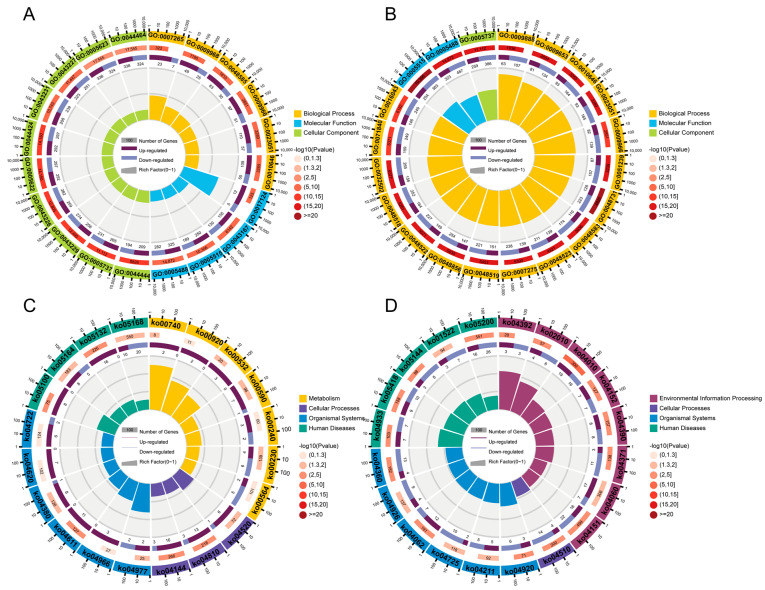
Unique DEGs caused by Pm HN01 and Pm HN02 enriched in different functional pathways. The GO enrichment results of unique DEGs in the Pm HN01 group (**A**) and Pm HN02 group (**B**), and the KEGG enrichment results of unique DEGs in the Pm HN01 group (**C**) and Pm HN02 group (**D**). Unique DEGs caused by Pm HN01 are mainly related to inflammation-related functions, but those caused by Pm HN02 are related to the Toll-like receptor signaling pathway.

**Figure 6 microorganisms-11-00400-f006:**
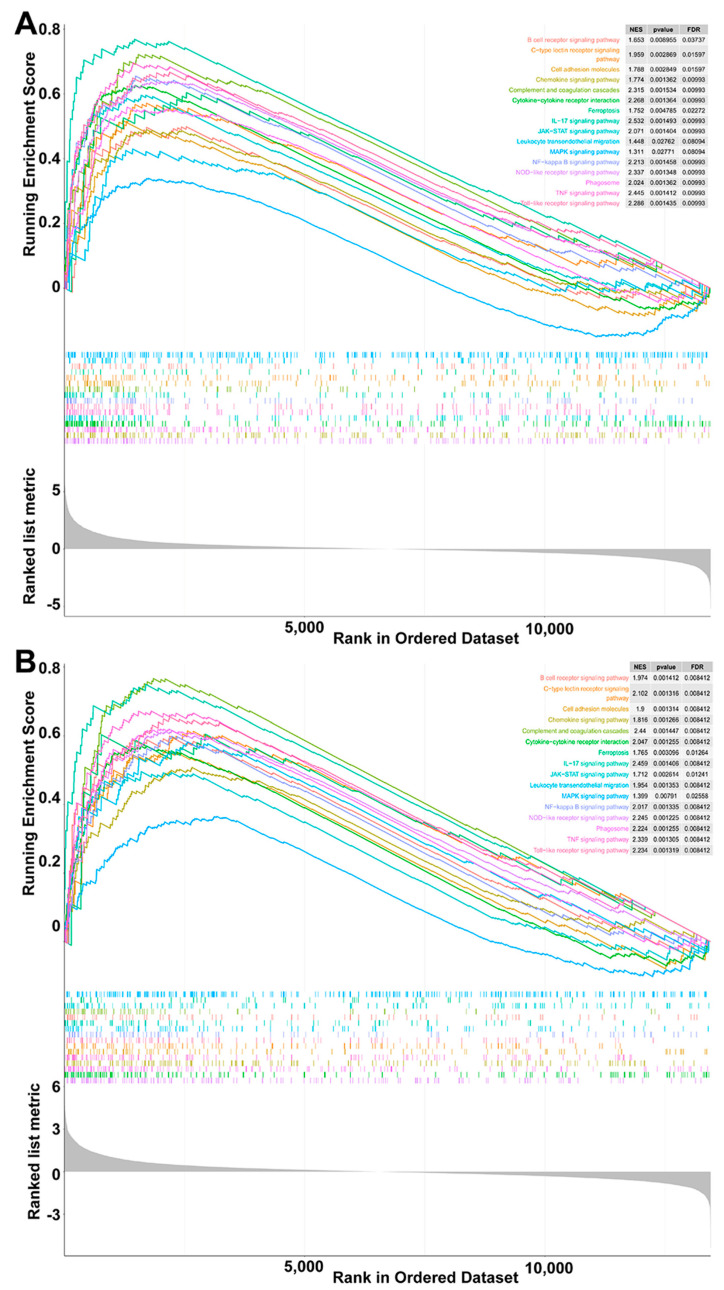
Sixteen pathways are activated in both the Pm HN01 and Pm HN02 groups. GSEA results of the Pm HN01 group (**A**) and Pm HN02 group (**B**).

**Figure 7 microorganisms-11-00400-f007:**
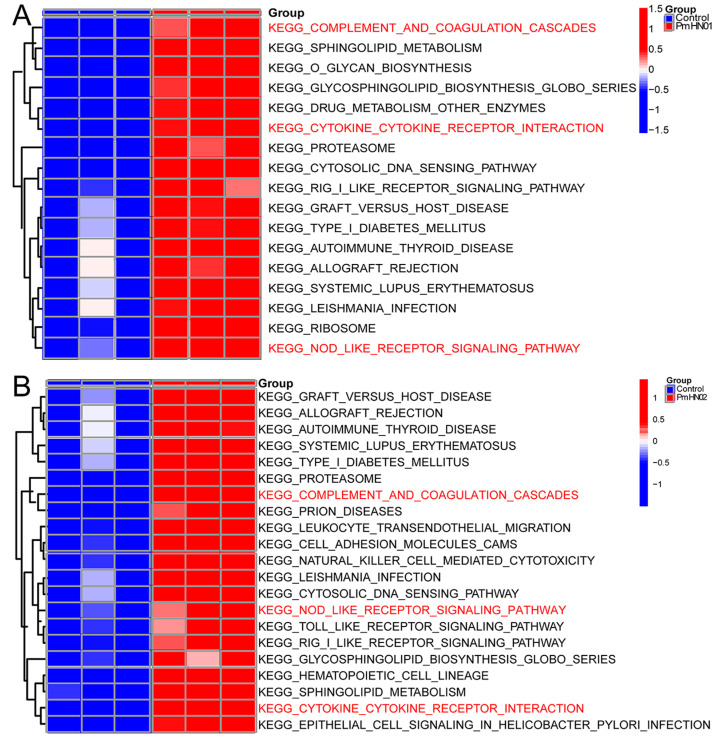
Toll-like receptor signaling pathway significantly activated by Pm HN02. GSVA for Pm HN01 (**A**) and Pm HN02 (**B**) groups. Three pathways (the red text) related to the innate immune system were significantly activated by Pm HN01 and Pm HN02. The Toll-like receptor signaling pathway was only significantly upregulated in the Pm HN02 group, with no significant difference in the Pm HN01 group.

**Figure 8 microorganisms-11-00400-f008:**
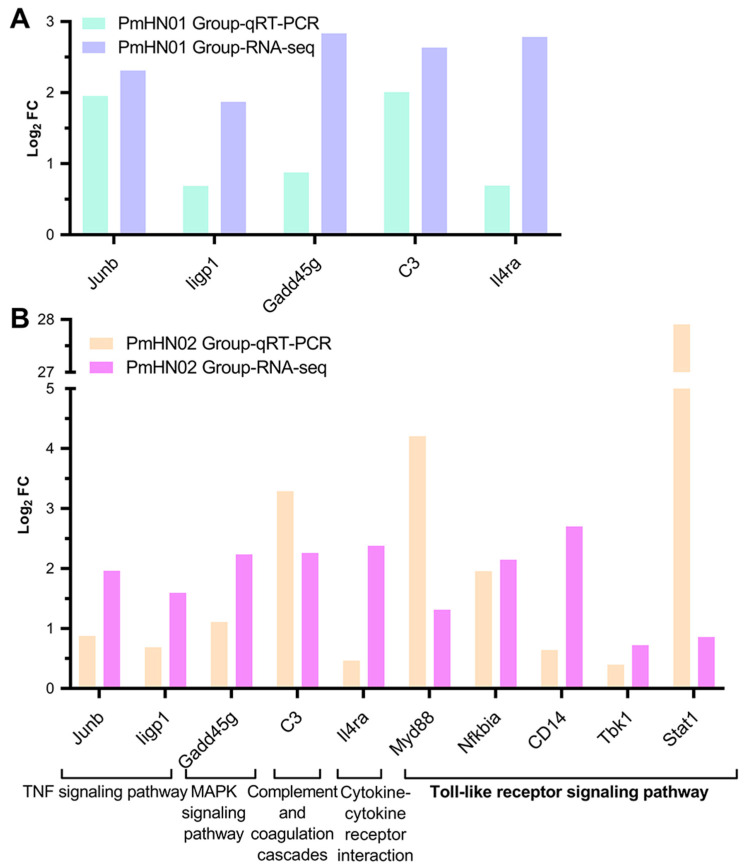
RT-qPCR validation. The result of comparing RT-qPCR with RNA sequencing based on log_2_ fold change. (**A**) is for Pm HN01 Group, and (**B**) is for Pm HN02 Group.

**Figure 9 microorganisms-11-00400-f009:**
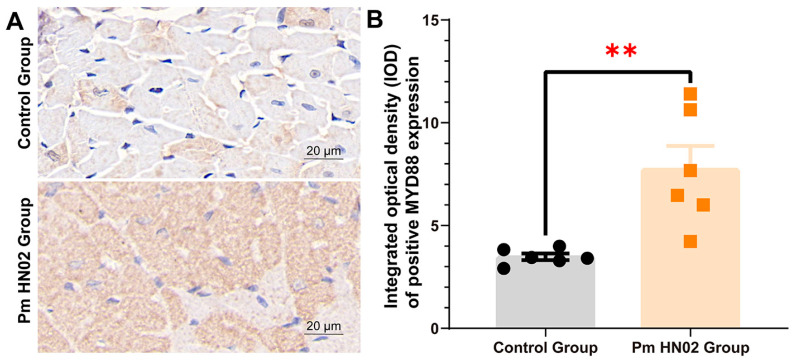
Hub protein related to Toll-like receptor signaling pathway in Pm HN02 group. (**A**) The expression of MyD88 in murine hearts shown by immunohistochemistry staining. (**B**) Comparing the expression levels of MyD88 in control group (*n* =  6) and Pm HN02 group (*n* =  6) murine hearts. ** *p* < 0.01 from two-tailed unpaired Student’s *t*-tests.

## Data Availability

The raw sequence data reported in this paper have been deposited in the Genome Sequence Archive in the National Genomics Data Center, China National Center for Bioinformation/Beijing Institute of Genomics, Chinese Academy of Sciences, under accession number CRA007332, which are publicly accessible at https://bigd.big.ac.cn/gsa.

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
