# Peer review of "Activation of MyD88-Dependent TLR Signaling Modulates Immune Response of the Mouse Heart during Pasteurella multocida Infection"

_microorganisms, 2023, doi:10.3390/microorganisms11020400_

Round 1

Reviewer 1 Report

The study is very well organised and methodologically correct. I have only some minor comments as follows:

I suggest to add “in mice” in the title

Line 61, …our colleagues (please add a reference)

Line 107, please add a reference or at least for the routine protocols 

Table 1. Please indicate more details in Table 1 by providing more columns such as amplicon length and gene name 

Author Response

 Thank you very much for your recognition, as well as your pertinent comments and suggestions on our article. Here, we have made the replies to the questions as following. Please, find attached my comments.

Reviewer 2 Report

Manuscript ID: microorganisms-2177210 “Activation of Myd88-Dependent TLRs participates Immune Response of the Heart during Pasteurella multocida Infection” by Qiaoyu Fu, Junming Jiang, Xubo Li, Zhe Zhai, Xuemei Wang, Chongrui Li, Qiaoling Chen, Churiga Man, Li Du, Fengyang Wang, and Si Chen

Pasteurellosis is an acute infection with a bacterial species of the genus Pasteurella, found in humans and animals. It is characterized by bronchopneumonia and septicemia. The organism invades the mucosa of the gastrointestinal or respiratory tract and causes localized areas of necrosis, hemorrhage, and thrombosis. The lungs and liver are frequent locations for the formation of microabscesses. Pasteurella may also infect the heart and has been reported as a rare cause of endocarditis.

Since not much is known about this, the manuscript's authors investigated the host immune response (on transcript level) of Pasteurella (P.) multocida-infected hearts. Therefore, Fu et al. established two murine infection models using P. multocida serotype A (Pm HN02) and D (Pm HN01) strains. The histopathological studies of the manuscript revealed inflammatory reactions and immune cell infiltration in epicardial/myocardial areas of investigated mouse hearts. Based on transcriptome sequencing of infected tissues, the authors performed functional enrichment analysis of DEGs and gene set enrichment/variation analysis. The findings suggest that both model groups significantly regulated different immune-relevant pathways (like the NLR receptor signaling pathway). Interestingly, the TLR signaling route was only activated considerably for the Pm HN02 and not for the Pm HN01 group. The observed up-regulation of TLR-hub protein MYD88 was further verified by immunohistochemistry. The authors propose that their study recovered critical pathways for the recognition and defense against P. multocida serotypes A and D in infected hearts providing a basis for future studies on the prevention, diagnosis, and treatment of Pasteurellosis.

P. multocida-infected hearts have been reported previously in a few studies and case reports. However, this kind of infection happens not very often. Thus, infective endocarditis (IE) is a rare manifestation of P. multocida infection. Therefore, it is not really clear to the reader of the manuscript why the authors put such a great effort (on a relatively rare phenomenon) into their studies. Most commonly, endocarditis caused by P. multocida affects prosthetic valves in immunocompromised patients. Some articles have reported P. multocida-induced endocarditis in native heart valves, but predisposing medical conditions were present in all patients. Nevertheless, it is conceivable that IE could present a diagnostic and therapeutic dilemma because of its rarity. Thus, the authors should provide more detail on their scientific intent and perspective for their experimental work. The few vague sentences in the introduction are not sufficient! 

Still, the work seems solidly done and has produced interesting data. From my point of view, the manuscript is worthy of publication after some necessary changes.

Points of critique:

·         However, one would like to see the illustrations/figures much larger, as the text within the illustrations is hard to read. This needs to be changed/improved (see figures 3-7).

·         The accuracy of the transcriptome sequencing results (RNAseq) of the Pm HN01 group should also be tested for 7+3 genes (as for the Pm HN02 group) by RT-qPCR.

·         Fig. 2 is entirely meaningless and should be removed from the manuscript.

·         The manuscript also contains some clerical errors. For example, in Fig. 8, the study refers to Pm HN02, whereas the accompanying legend refers to Pm HN01. Which of the two pieces of information/statements is correct in Fig. 8?

·         The authors should replace "IOD of Positive" in Fig. 8B with "Integral optical density (IOD) of positive MYD88 expression".

·         The experiment depicted in Fig. 8A/B should be performed in parallel for hearts infected with Pm HN01 (serotype D) or Pm HN02 (serotype A). Only in this way is the experiment meaningful.

Author Response

We have carefully reviewed the comments and suggestions. Here, we have made the replies to the questions as following. Please, find attached my comments.
